# Masked Autoregressive Flow for Density Estimation

**George Papamakarios**
University of Edinburgh
g.papamakarios@ed.ac.uk

**Theo Pavlakou**
University of Edinburgh
theo.pavlakou@ed.ac.uk

**Iain Murray**
University of Edinburgh
i.murray@ed.ac.uk

## Abstract

Autoregressive models are among the best performing neural density estimators. We describe an approach for increasing the flexibility of an autoregressive model, based on modelling the random numbers that the model uses internally when generating data. By constructing a stack of autoregressive models, each modelling the random numbers of the next model in the stack, we obtain a type of normalizing flow suitable for density estimation, which we call Masked Autoregressive Flow. This type of flow is closely related to Inverse Autoregressive Flow and is a generalization of Real NVP. Masked Autoregressive Flow achieves state-of-the-art performance in a range of general-purpose density estimation tasks.

## 1 Introduction

The joint density $p(\mathbf{x})$ of a set of variables $\mathbf{x}$ is a central object of interest in machine learning. Being able to access and manipulate $p(\mathbf{x})$ enables a wide range of tasks to be performed, such as inference, prediction, data completion and data generation. As such, the problem of estimating $p(\mathbf{x})$ from a set of examples $\{\mathbf{x}_n\}$ is at the core of probabilistic unsupervised learning and generative modelling.

In recent years, using neural networks for density estimation has been particularly successful. Combining the flexibility and learning capacity of neural networks with prior knowledge about the structure of data to be modelled has led to impressive results in modelling natural images [4, 30, 37, 38] and audio data [34, 36]. State-of-the-art neural density estimators have also been used for likelihood-free inference from simulated data [21, 23], variational inference [13, 24], and as surrogates for maximum entropy models [19].

Neural density estimators differ from other approaches to generative modelling—such as variational autoencoders [12, 25] and generative adversarial networks [7]—in that they readily provide exact density evaluations. As such, they are more suitable in applications where the focus is on explicitly evaluating densities, rather than generating synthetic data. For instance, density estimators can learn suitable priors for data from large unlabelled datasets, for use in standard Bayesian inference [39]. In simulation-based likelihood-free inference, conditional density estimators can learn models for the likelihood [5] or the posterior [23] from simulated data. Density estimators can learn effective proposals for importance sampling [22] or sequential Monte Carlo [8, 21]; such proposals can be used in probabilistic programming environments to speed up inference [15, 16]. Finally, conditional density estimators can be used as flexible inference networks for amortized variational inference and as part of variational autoencoders [12, 25].

A challenge in neural density estimation is to construct models that are flexible enough to represent complex densities, but have tractable density functions and learning algorithms. There are mainly two families of neural density estimators that are both flexible and tractable: *autoregressive models* [35] and *normalizing flows* [24]. Autoregressive models decompose the joint density as a product of conditionals, and model each conditional in turn. Normalizing flows transform a base density (e.g. a standard Gaussian) into the target density by an invertible transformation with tractable Jacobian.

Our starting point is the realization (as pointed out by Kingma et al. [13]) that autoregressive models, when used to generate data, correspond to a differentiable transformation of an external source of randomness (typically obtained by random number generators). This transformation has a tractable Jacobian by design, and for certain autoregressive models it is also invertible, hence it precisely corresponds to a normalizing flow. Viewing an autoregressive model as a normalizing flow opens the possibility of increasing its flexibility by stacking multiple models of the same type, by having each model provide the source of randomness for the next model in the stack. The resulting stack of models is a normalizing flow that is more flexible than the original model, and that remains tractable.

In this paper we present *Masked Autoregressive Flow* (MAF), which is a particular implementation of the above normalizing flow that uses the Masked Autoencoder for Distribution Estimation (MADE) [6] as a building block. The use of MADE enables density evaluations without the sequential loop that is typical of autoregressive models, and thus makes MAF fast to evaluate and train on parallel computing architectures such as Graphics Processing Units (GPUs). We show a close theoretical connection between MAF and Inverse Autoregressive Flow (IAF) [13], which has been designed for variational inference instead of density estimation, and show that both correspond to generalizations of the successful Real NVP [4]. We experimentally evaluate MAF on a wide range of datasets, and we demonstrate that (a) MAF outperforms Real NVP on general-purpose density estimation, and (b) a conditional version of MAF achieves close to state-of-the-art performance on conditional image modelling even with a general-purpose architecture.

## 2 Background

### 2.1 Autoregressive density estimation

Using the chain rule of probability, any joint density $p(\mathbf{x})$ can be decomposed into a product of one-dimensional conditionals as $p(\mathbf{x}) = \prod_i p(x_i \,|\, \mathbf{x}_{1:i-1})$. Autoregressive density estimators [35] model each conditional $p(x_i \,|\, \mathbf{x}_{1:i-1})$ as a parametric density, whose parameters are a function of a hidden state $\mathbf{h}_i$. In recurrent architectures, $\mathbf{h}_i$ is a function of the previous hidden state $\mathbf{h}_{i-1}$ and the $i^{\text{th}}$ input variable $x_i$. The Real-valued Neural Autoregressive Density Estimator (RNADE) [32] uses mixtures of Gaussian or Laplace densities for modelling the conditionals, and a simple linear rule for updating the hidden state. More flexible approaches for updating the hidden state are based on Long Short-Term Memory recurrent neural networks [30, 38].

A drawback of autoregressive models is that they are sensitive to the order of the variables. For example, the order of the variables matters when learning the density of Figure 1a if we assume a model with Gaussian conditionals. As Figure 1b shows, a model with order $(x_1, x_2)$ cannot learn this density, even though the same model with order $(x_2, x_1)$ can represent it perfectly. In practice is it hard to know which of the factorially many orders is the most suitable for the task at hand. Autoregressive models that are trained to work with an order chosen at random have been developed, and the predictions from different orders can then be combined in an ensemble [6, 33]. Our approach (Section 3) can use a different order in each layer, and using random orders would also be possible.

Straightforward recurrent autoregressive models would update a hidden state sequentially for every variable, requiring $D$ sequential computations to compute the probability $p(\mathbf{x})$ of a $D$-dimensional vector, which is not well-suited for computation on parallel architectures such as GPUs. One way to enable parallel computation is to start with a fully-connected model with $D$ inputs and $D$ outputs, and drop out connections in order to ensure that output $i$ will only be connected to inputs $1, 2, \ldots, i-1$. Output $i$ can then be interpreted as computing the parameters of the $i^{\text{th}}$ conditional $p(x_i \,|\, \mathbf{x}_{1:i-1})$. By construction, the resulting model will satisfy the autoregressive property, and at the same time it will be able to calculate $p(\mathbf{x})$ efficiently on a GPU. An example of this approach is the Masked Autoencoder for Distribution Estimation (MADE) [6], which drops out connections by multiplying the weight matrices of a fully-connected autoencoder with binary masks. Other mechanisms for dropping out connections include masked convolutions [38] and causal convolutions [36].

### 2.2 Normalizing flows

A normalizing flow [24] represents $p(\mathbf{x})$ as an invertible differentiable transformation $f$ of a base density $\pi_u(\mathbf{u})$. That is, $\mathbf{x} = f(\mathbf{u})$ where $\mathbf{u} \sim \pi_u(\mathbf{u})$. The base density $\pi_u(\mathbf{u})$ is chosen such that it can be easily evaluated for any input $\mathbf{u}$ (a common choice for $\pi_u(\mathbf{u})$ is a standard Gaussian). Under

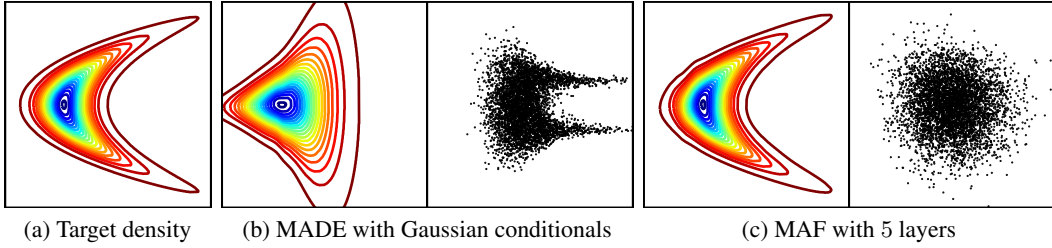

(a) Target density     (b) MADE with Gaussian conditionals     (c) MAF with 5 layers

Figure 1: **(a)** The density to be learnt, defined as $p(x_1, x_2) = \mathcal{N}(x_2 \,|\, 0, 4)\mathcal{N}\big(x_1 \,|\, \frac{1}{4}x_2^2, 1\big)$. **(b)** The density learnt by a MADE with order $(x_1, x_2)$ and Gaussian conditionals. Scatter plot shows the train data transformed into random numbers $\mathbf{u}$; the non-Gaussian distribution indicates that the model is a poor fit. **(c)** Learnt density and transformed train data of a 5 layer MAF with the same order $(x_1, x_2)$.

the invertibility assumption for $f$, the density $p(\mathbf{x})$ can be calculated as

$$p(\mathbf{x}) = \pi_u\big(f^{-1}(\mathbf{x})\big) \left| \det\left(\frac{\partial f^{-1}}{\partial \mathbf{x}}\right) \right|. \tag{1}$$

In order for Equation (1) to be tractable, the transformation $f$ must be constructed such that (a) it is easy to invert, and (b) the determinant of its Jacobian is easy to compute. An important point is that if transformations $f_1$ and $f_2$ have the above properties, then their composition $f_1 \circ f_2$ also has these properties. In other words, the transformation $f$ can be made deeper by composing multiple instances of it, and the result will still be a valid normalizing flow.

There have been various approaches in developing normalizing flows. An early example is Gaussianization [2], which is based on successive application of independent component analysis. Enforcing invertibility with nonsingular weight matrices has been proposed [1, 26], however in such approaches calculating the determinant of the Jacobian scales cubically with data dimensionality in general. Planar/radial flows [24] and Inverse Autoregressive Flow (IAF) [13] are models whose Jacobian is tractable by design. However, they were developed primarily for variational inference and are not well-suited for density estimation, as they can only efficiently calculate the density of their own samples and not of externally provided datapoints. The Non-linear Independent Components Estimator (NICE) [3] and its successor Real NVP [4] have a tractable Jacobian and are also suitable for density estimation. IAF, NICE and Real NVP are discussed in more detail in Section 3.

## 3   Masked Autoregressive Flow

### 3.1   Autoregressive models as normalizing flows

Consider an autoregressive model whose conditionals are parameterized as single Gaussians. That is, the $i^{\text{th}}$ conditional is given by

$$p(x_i \,|\, \mathbf{x}_{1:i-1}) = \mathcal{N}\big(x_i \,|\, \mu_i, (\exp \alpha_i)^2\big) \quad \text{where} \quad \mu_i = f_{\mu_i}(\mathbf{x}_{1:i-1}) \quad \text{and} \quad \alpha_i = f_{\alpha_i}(\mathbf{x}_{1:i-1}). \tag{2}$$

In the above, $f_{\mu_i}$ and $f_{\alpha_i}$ are unconstrained scalar functions that compute the mean and log standard deviation of the $i^{\text{th}}$ conditional given all previous variables. We can generate data from the above model using the following recursion:

$$x_i = u_i \exp \alpha_i + \mu_i \quad \text{where} \quad \mu_i = f_{\mu_i}(\mathbf{x}_{1:i-1}), \;\; \alpha_i = f_{\alpha_i}(\mathbf{x}_{1:i-1}) \;\; \text{and} \;\; u_i \sim \mathcal{N}(0,1). \tag{3}$$

In the above, $\mathbf{u} = (u_1, u_2, \ldots, u_I)$ is the vector of random numbers the model uses internally to generate data, typically by making calls to a random number generator often called `randn()`.

Equation (3) provides an alternative characterization of the autoregressive model as a transformation $f$ from the space of random numbers $\mathbf{u}$ to the space of data $\mathbf{x}$. That is, we can express the model as $\mathbf{x} = f(\mathbf{u})$ where $\mathbf{u} \sim \mathcal{N}(\mathbf{0}, \mathbf{I})$. By construction, $f$ is easily invertible. Given a datapoint $\mathbf{x}$, the random numbers $\mathbf{u}$ that were used to generate it are obtained by the following recursion:

$$u_i = (x_i - \mu_i) \exp(-\alpha_i) \quad \text{where} \quad \mu_i = f_{\mu_i}(\mathbf{x}_{1:i-1}) \quad \text{and} \quad \alpha_i = f_{\alpha_i}(\mathbf{x}_{1:i-1}). \tag{4}$$

Due to the autoregressive structure, the Jacobian of $f^{-1}$ is triangular by design, hence its absolute determinant can be easily obtained as follows:

$$\left| \det\left(\frac{\partial f^{-1}}{\partial \mathbf{x}}\right) \right| = \exp\left(-\sum_i \alpha_i\right) \quad \text{where} \quad \alpha_i = f_{\alpha_i}(\mathbf{x}_{1:i-1}). \tag{5}$$

It follows that the autoregressive model can be equivalently interpreted as a normalizing flow, whose density $p(\mathbf{x})$ can be obtained by substituting Equations (4) and (5) into Equation (1). This observation was first pointed out by Kingma et al. [13].

A useful diagnostic for assessing whether an autoregressive model of the above type fits the target density well is to transform the train data $\{\mathbf{x}_n\}$ into corresponding random numbers $\{\mathbf{u}_n\}$ using Equation (4), and assess whether the $u_i$'s come from independent standard normals. If the $u_i$'s do not seem to come from independent standard normals, this is evidence that the model is a bad fit. For instance, Figure 1b shows that the scatter plot of the random numbers associated with the train data can look significantly non-Gaussian if the model fits the target density poorly.

Here we interpret autoregressive models as a flow, and improve the model fit by stacking multiple instances of the model into a deeper flow. Given autoregressive models $M_1, M_2, \ldots, M_K$, we model the density of the random numbers $\mathbf{u}_1$ of $M_1$ with $M_2$, model the random numbers $\mathbf{u}_2$ of $M_2$ with $M_3$ and so on, finally modelling the random numbers $\mathbf{u}_K$ of $M_K$ with a standard Gaussian. This stacking adds flexibility: for example, Figure 1c demonstrates that a flow of 5 autoregressive models is able to learn multimodal conditionals, even though each model has unimodal conditionals. Stacking has previously been used in a similar way to improve model fit of deep belief nets [9] and deep mixtures of factor analyzers [28].

We choose to implement the set of functions $\{f_{\mu_i}, f_{\alpha_i}\}$ with *masking*, following the approach used by MADE [6]. MADE is a feedforward network that takes $\mathbf{x}$ as input and outputs $\mu_i$ and $\alpha_i$ for all $i$ with a single forward pass. The autoregressive property is enforced by multiplying the weight matrices of MADE with suitably constructed binary masks. In other words, we use MADE with Gaussian conditionals as the building layer of our flow. The benefit of using masking is that it enables transforming from data $\mathbf{x}$ to random numbers $\mathbf{u}$ and thus calculating $p(\mathbf{x})$ in one forward pass through the flow, thus eliminating the need for sequential recursion as in Equation (4). We call this implementation of stacking MADEs into a flow *Masked Autoregressive Flow* (MAF).

### 3.2 Relationship with Inverse Autoregressive Flow

Like MAF, Inverse Autoregressive Flow (IAF) [13] is a normalizing flow which uses MADE as its component layer. Each layer of IAF is defined by the following recursion:

$$x_i = u_i \exp \alpha_i + \mu_i \quad \text{where} \quad \mu_i = f_{\mu_i}(\mathbf{u}_{1:i-1}) \text{ and } \alpha_i = f_{\alpha_i}(\mathbf{u}_{1:i-1}). \tag{6}$$

Similarly to MAF, functions $\{f_{\mu_i}, f_{\alpha_i}\}$ are computed using a MADE with Gaussian conditionals. The difference is architectural: in MAF $\mu_i$ and $\alpha_i$ are directly computed from previous *data variables* $\mathbf{x}_{1:i-1}$, whereas in IAF $\mu_i$ and $\alpha_i$ are directly computed from previous *random numbers* $\mathbf{u}_{1:i-1}$.

The consequence of the above is that MAF and IAF are different models with different computational trade-offs. MAF is capable of calculating the density $p(\mathbf{x})$ of *any* datapoint $\mathbf{x}$ in one pass through the model, however sampling from it requires performing $D$ sequential passes (where $D$ is the dimensionality of $\mathbf{x}$). In contrast, IAF can generate samples and calculate their density with one pass, however calculating the density $p(\mathbf{x})$ of an externally provided datapoint $\mathbf{x}$ requires $D$ passes to find the random numbers $\mathbf{u}$ associated with $\mathbf{x}$. Hence, the design choice of whether to connect $\mu_i$ and $\alpha_i$ directly to $\mathbf{x}_{1:i-1}$ (obtaining MAF) or to $\mathbf{u}_{1:i-1}$ (obtaining IAF) depends on the intended usage. IAF is suitable as a recognition model for stochastic variational inference [12, 25], where it only ever needs to calculate the density of its own samples. In contrast, MAF is more suitable for density estimation, because each example requires only one pass through the model whereas IAF requires $D$.

A theoretical equivalence between MAF and IAF is that training a MAF with maximum likelihood corresponds to fitting an implicit IAF to the base density with stochastic variational inference. Let $\pi_x(\mathbf{x})$ be the data density we wish to learn, $\pi_u(\mathbf{u})$ be the base density, and $f$ be the transformation from $\mathbf{u}$ to $\mathbf{x}$ as implemented by MAF. The density defined by MAF (with added subscript $x$ for disambiguation) is

$$p_x(\mathbf{x}) = \pi_u\left(f^{-1}(\mathbf{x})\right) \left| \det\left(\frac{\partial f^{-1}}{\partial \mathbf{x}}\right) \right|. \tag{7}$$

The inverse transformation $f^{-1}$ from $\mathbf{x}$ to $\mathbf{u}$ can be seen as describing an implicit IAF with base density $\pi_x(\mathbf{x})$, which defines the following implicit density over the $\mathbf{u}$ space:

$$p_u(\mathbf{u}) = \pi_x(f(\mathbf{u})) \left| \det\left( \frac{\partial f}{\partial \mathbf{u}} \right) \right|. \tag{8}$$

Training MAF by maximizing the total log likelihood $\sum_n \log p(\mathbf{x}_n)$ on train data $\{\mathbf{x}_n\}$ corresponds to fitting $p_x(\mathbf{x})$ to $\pi_x(\mathbf{x})$ by stochastically minimizing $D_{\mathrm{KL}}(\pi_x(\mathbf{x}) \,\|\, p_x(\mathbf{x}))$. In Section A of the supplementary material, we show that

$$D_{\mathrm{KL}}(\pi_x(\mathbf{x}) \,\|\, p_x(\mathbf{x})) = D_{\mathrm{KL}}(p_u(\mathbf{u}) \,\|\, \pi_u(\mathbf{u})). \tag{9}$$

Hence, stochastically minimizing $D_{\mathrm{KL}}(\pi_x(\mathbf{x}) \,\|\, p_x(\mathbf{x}))$ is equivalent to fitting $p_u(\mathbf{u})$ to $\pi_u(\mathbf{u})$ by minimizing $D_{\mathrm{KL}}(p_u(\mathbf{u}) \,\|\, \pi_u(\mathbf{u}))$. Since the latter is the loss function used in variational inference, and $p_u(\mathbf{u})$ can be seen as an IAF with base density $\pi_x(\mathbf{x})$ and transformation $f^{-1}$, it follows that training MAF as a density estimator of $\pi_x(\mathbf{x})$ is equivalent to performing stochastic variational inference with an implicit IAF, where the posterior is taken to be the base density $\pi_u(\mathbf{u})$ and the transformation $f^{-1}$ implements the reparameterization trick [12, 25]. This argument is presented in more detail in Section A of the supplementary material.

### 3.3 Relationship with Real NVP

Real NVP [4] (NVP stands for Non Volume Preserving) is a normalizing flow obtained by stacking *coupling layers*. A coupling layer is an invertible transformation $f$ from random numbers $\mathbf{u}$ to data $\mathbf{x}$ with a tractable Jacobian, defined by

$$\begin{aligned} \mathbf{x}_{1:d} &= \mathbf{u}_{1:d} \\ \mathbf{x}_{d+1:D} &= \mathbf{u}_{d+1:D} \odot \exp \boldsymbol{\alpha} + \boldsymbol{\mu} \end{aligned} \quad \text{where} \quad \begin{aligned} \boldsymbol{\mu} &= f_\mu(\mathbf{u}_{1:d}) \\ \boldsymbol{\alpha} &= f_\alpha(\mathbf{u}_{1:d}). \end{aligned} \tag{10}$$

In the above, $\odot$ denotes elementwise multiplication, and the $\exp$ is applied to each element of $\boldsymbol{\alpha}$. The transformation copies the first $d$ elements, and scales and shifts the remaining $D-d$ elements, with the amount of scaling and shifting being a function of the first $d$ elements. When stacking coupling layers into a flow, the elements are permuted across layers so that a different set of elements is copied each time. A special case of the coupling layer where $\boldsymbol{\alpha} = \mathbf{0}$ is used by NICE [3].

We can see that the coupling layer is a special case of both the autoregressive transformation used by MAF in Equation (3), and the autoregressive transformation used by IAF in Equation (6). Indeed, we can recover the coupling layer from the autoregressive transformation of MAF by setting $\mu_i = \alpha_i = 0$ for $i \leq d$ and making $\mu_i$ and $\alpha_i$ functions of only $\mathbf{x}_{1:d}$ for $i > d$ (for IAF we need to make $\mu_i$ and $\alpha_i$ functions of $\mathbf{u}_{1:d}$ instead for $i > d$). In other words, both MAF and IAF can be seen as more flexible (but different) generalizations of Real NVP, where each element is individually scaled and shifted as a function of all previous elements. The advantage of Real NVP compared to MAF and IAF is that it can both generate data and estimate densities with one forward pass only, whereas MAF would need $D$ passes to generate data and IAF would need $D$ passes to estimate densities.

### 3.4 Conditional MAF

Given a set of example pairs $\{(\mathbf{x}_n, \mathbf{y}_n)\}$, conditional density estimation is the task of estimating the conditional density $p(\mathbf{x} \,|\, \mathbf{y})$. Autoregressive modelling extends naturally to conditional density estimation. Each term in the chain rule of probability can be conditioned on side-information $\mathbf{y}$, decomposing any conditional density as $p(\mathbf{x} \,|\, \mathbf{y}) = \prod_i p(x_i \,|\, \mathbf{x}_{1:i-1}, \mathbf{y})$. Therefore, we can turn any unconditional autoregressive model into a conditional one by augmenting its set of input variables with $\mathbf{y}$ and only modelling the conditionals that correspond to $\mathbf{x}$. Any order of the variables can be chosen, as long as $\mathbf{y}$ comes before $\mathbf{x}$. In masked autoregressive models, no connections need to be dropped from the $\mathbf{y}$ inputs to the rest of the network.

We can implement a conditional version of MAF by stacking MADEs that were made conditional using the above strategy. That is, in a conditional MAF, the vector $\mathbf{y}$ becomes an additional input for every layer. As a special case of MAF, Real NVP can be made conditional in the same way. In Section 4, we show that conditional MAF significantly outperforms unconditional MAF when conditional information (such as data labels) is available. In our experiments, MAF was able to benefit from conditioning considerably more than MADE and Real NVP.

## 4 Experiments

### 4.1 Implementation and setup

We systematically evaluate three types of density estimator (MADE, Real NVP and MAF) in terms of density estimation performance on a variety of datasets. Code for reproducing our experiments (which uses Theano [29]) can be found at `https://github.com/gpapamak/maf`.

**MADE.** We consider two versions: (a) a MADE with Gaussian conditionals, denoted simply by MADE, and (b) a MADE whose conditionals are each parameterized as a mixture of $C$ Gaussians, denoted by MADE MoG. We used $C = 10$ in all our experiments. MADE can be seen either as a MADE MoG with $C = 1$, or as a MAF with only one autoregressive layer. Adding more Gaussian components per conditional or stacking MADEs to form a MAF are two alternative ways of increasing the flexibility of MADE, which we are interested in comparing.

**Real NVP.** We consider a general-purpose implementation of the coupling layer, which uses two feedforward neural networks, implementing the scaling function $f_\alpha$ and the shifting function $f_\mu$ respectively. Both networks have the same architecture, except that $f_\alpha$ has hyperbolic tangent hidden units, whereas $f_\mu$ has rectified linear hidden units (we found this combination to perform best). Both networks have a linear output. We consider Real NVPs with either 5 or 10 coupling layers, denoted by Real NVP (5) and Real NVP (10) respectively, and in both cases the base density is a standard Gaussian. Successive coupling layers alternate between (a) copying the odd-indexed variables and transforming the even-indexed variables, and (b) copying the even-indexed variables and transforming the odd-indexed variables. It is important to clarify that this is a general-purpose implementation of Real NVP which is different and thus not comparable to its original version [4], which was designed specifically for image data. Here we are interested in comparing coupling layers with autoregressive layers as building blocks of normalizing flows for general-purpose density estimation tasks, and our design of Real NVP is such that a fair comparison between the two can be made.

**MAF.** We consider three versions: (a) a MAF with 5 autoregressive layers and a standard Gaussian as a base density $\pi_u(\mathbf{u})$, denoted by MAF (5), (b) a MAF with 10 autoregressive layers and a standard Gaussian as a base density, denoted by MAF (10), and (c) a MAF with 5 autoregressive layers and a MADE MoG with $C = 10$ Gaussian components as a base density, denoted by MAF MoG (5). MAF MoG (5) can be thought of as a MAF (5) stacked on top of a MADE MoG and trained jointly with it.

In all experiments, MADE and MADE MoG order the inputs using the order that comes with the dataset by default; no alternative orders were considered. MAF uses the default order for the first autoregressive layer (i.e. the layer that directly models the data) and reverses the order for each successive layer (the same was done for IAF by Kingma et al. [13]).

MADE, MADE MoG and each layer in MAF is a feedforward neural network with masked weight matrices, such that the autoregressive property holds. The procedure for designing the masks (due to Germain et al. [6]) is as follows. Each input or hidden unit is assigned a degree, which is an integer ranging from 1 to $D$, where $D$ is the data dimensionality. The degree of an input is taken to be its index in the order. The $D$ outputs have degrees that sequentially range from 0 to $D-1$. A unit is allowed to receive input only from units with lower or equal degree, which enforces the autoregressive property. In order for output $i$ to be connected to all inputs with degree less than $i$, and thus make sure that no conditional independences are introduced, it is both necessary and sufficient that every hidden layer contains every degree. In all experiments except for CIFAR-10, we sequentially assign degrees within each hidden layer and use enough hidden units to make sure that all degrees appear. Because CIFAR-10 is high-dimensional, we used fewer hidden units than inputs and assigned degrees to hidden units uniformly at random (as was done by Germain et al. [6]).

We added batch normalization [10] after each coupling layer in Real NVP and after each autoregressive layer in MAF. Batch normalization is an elementwise scaling and shifting, which is easily invertible and has a tractable Jacobian, and thus it is suitable for use in a normalizing flow. We found that batch normalization in Real NVP and MAF reduces training time, increases stability during training and improves performance (as observed by Dinh et al. [4] for Real NVP). Section B of the supplementary material discusses our implementation of batch normalization and its use in normalizing flows.

All models were trained with the Adam optimizer [11], using a minibatch size of 100, and a step size of $10^{-3}$ for MADE and MADE MoG, and of $10^{-4}$ for Real NVP and MAF. A small amount of $\ell_2$

Table 1: Average test log likelihood (in nats) for unconditional density estimation. The best performing model for each dataset is shown in bold (multiple models are highlighted if the difference is not statistically significant according to a paired $t$-test). Error bars correspond to 2 standard deviations.

|  | POWER | GAS | HEPMASS | MINIBOONE | BSDS300 |
|---|---|---|---|---|---|
| Gaussian | $-7.74 \pm 0.02$ | $-3.58 \pm 0.75$ | $-27.93 \pm 0.02$ | $-37.24 \pm 1.07$ | $96.67 \pm 0.25$ |
| MADE | $-3.08 \pm 0.03$ | $3.56 \pm 0.04$ | $-20.98 \pm 0.02$ | $-15.59 \pm 0.50$ | $148.85 \pm 0.28$ |
| MADE MoG | $\mathbf{0.40 \pm 0.01}$ | $8.47 \pm 0.02$ | $\mathbf{-15.15 \pm 0.02}$ | $-12.27 \pm 0.47$ | $153.71 \pm 0.28$ |
| Real NVP (5) | $-0.02 \pm 0.01$ | $4.78 \pm 1.80$ | $-19.62 \pm 0.02$ | $-13.55 \pm 0.49$ | $152.97 \pm 0.28$ |
| Real NVP (10) | $0.17 \pm 0.01$ | $8.33 \pm 0.14$ | $-18.71 \pm 0.02$ | $-13.84 \pm 0.52$ | $153.28 \pm 1.78$ |
| MAF (5) | $0.14 \pm 0.01$ | $9.07 \pm 0.02$ | $-17.70 \pm 0.02$ | $\mathbf{-11.75 \pm 0.44}$ | $155.69 \pm 0.28$ |
| MAF (10) | $0.24 \pm 0.01$ | $\mathbf{10.08 \pm 0.02}$ | $-17.73 \pm 0.02$ | $-12.24 \pm 0.45$ | $154.93 \pm 0.28$ |
| MAF MoG (5) | $0.30 \pm 0.01$ | $9.59 \pm 0.02$ | $-17.39 \pm 0.02$ | $\mathbf{-11.68 \pm 0.44}$ | $\mathbf{156.36 \pm 0.28}$ |

regularization was added, with coefficient $10^{-6}$. Each model was trained with early stopping until no improvement occurred for 30 consecutive epochs on the validation set. For each model, we selected the number of hidden layers and number of hidden units based on validation performance (we gave the same options to all models), as described in Section D of the supplementary material.

## 4.2 Unconditional density estimation

Following Uria et al. [32], we perform unconditional density estimation on four UCI datasets (POWER, GAS, HEPMASS, MINIBOONE) and on a dataset of natural image patches (BSDS300).

**UCI datasets**. These datasets were taken from the UCI machine learning repository [18]. We selected different datasets than Uria et al. [32], because the ones they used were much smaller, resulting in an expensive cross-validation procedure involving a separate hyperparameter search for each fold. However, our data preprocessing follows Uria et al. [32]. The sample mean was subtracted from the data and each feature was divided by its sample standard deviation. Discrete-valued attributes were eliminated, as well as every attribute with a Pearson correlation coefficient greater than $0.98$. These procedures are meant to avoid trivial high densities, which would make the comparison between approaches hard to interpret. Section D of the supplementary material gives more details about the UCI datasets and the individual preprocessing done on each of them.

**Image patches**. This dataset was obtained by extracting random $8 \times 8$ monochrome patches from the BSDS300 dataset of natural images [20]. We used the same preprocessing as by Uria et al. [32]. Uniform noise was added to dequantize pixel values, which was then rescaled to be in the range $[0, 1]$. The mean pixel value was subtracted from each patch, and the bottom-right pixel was discarded.

Table 1 shows the performance of each model on each dataset. A Gaussian fitted to the train data is reported as a baseline. We can see that on 3 out of 5 datasets MAF is the best performing model, with MADE MoG being the best performing model on the other 2. On all datasets, MAF outperforms Real NVP. For the MINIBOONE dataset, due to overlapping error bars, a pairwise comparison was done to determine which model performs the best, the results of which are reported in Section E of the supplementary material. MAF MoG (5) achieves the best reported result on BSDS300 for a single model with 156.36 nats, followed by Deep RNADE [33] with 155.2. An ensemble of 32 Deep RNADEs was reported to achieve 157.0 nats [33]. The UCI datasets were used for the first time in the literature for density estimation, so no comparison with existing work can be made yet.

## 4.3 Conditional density estimation

For conditional density estimation, we used the MNIST dataset of handwritten digits [17] and the CIFAR-10 dataset of natural images [14]. In both datasets, each datapoint comes from one of 10 distinct classes. We represent the class label as a 10-dimensional, one-hot encoded vector $\mathbf{y}$, and we model the density $p(\mathbf{x} \mid \mathbf{y})$, where $\mathbf{x}$ represents an image. At test time, we evaluate the probability of a test image $\mathbf{x}$ by $p(\mathbf{x}) = \sum_{\mathbf{y}} p(\mathbf{x} \mid \mathbf{y}) p(\mathbf{y})$, where $p(\mathbf{y}) = \frac{1}{10}$ is a uniform prior over the labels. For comparison, we also train every model as an unconditional density estimator and report both results.

Table 2: Average test log likelihood (in nats) for conditional density estimation. The best performing model for each dataset is shown in bold. Error bars correspond to 2 standard deviations.

| | MNIST | | CIFAR-10 | |
|---|---|---|---|---|
| | unconditional | conditional | unconditional | conditional |
| Gaussian | $-1366.9 \pm 1.4$ | $-1344.7 \pm 1.8$ | $2367 \pm 29$ | $2030 \pm 41$ |
| MADE | $-1380.8 \pm 4.8$ | $-1361.9 \pm 1.9$ | $147 \pm 20$ | $187 \pm 20$ |
| MADE MoG | $\mathbf{-1038.5 \pm 1.8}$ | $-1030.3 \pm 1.7$ | $-397 \pm 21$ | $-119 \pm 20$ |
| Real NVP (5) | $-1323.2 \pm 6.6$ | $-1326.3 \pm 5.8$ | $2576 \pm 27$ | $2642 \pm 26$ |
| Real NVP (10) | $-1370.7 \pm 10.1$ | $-1371.3 \pm 43.9$ | $2568 \pm 26$ | $2475 \pm 25$ |
| MAF (5) | $-1300.5 \pm 1.7$ | $\mathbf{-591.7 \pm 1.7}$ | $2936 \pm 27$ | $5797 \pm 26$ |
| MAF (10) | $-1313.1 \pm 2.0$ | $-605.6 \pm 1.8$ | $\mathbf{3049 \pm 26}$ | $\mathbf{5872 \pm 26}$ |
| MAF MoG (5) | $-1100.3 \pm 1.6$ | $-1092.3 \pm 1.7$ | $2911 \pm 26$ | $2936 \pm 26$ |

For both MNIST and CIFAR-10, we use the same preprocessing as by Dinh et al. [4]. We dequantize pixel values by adding uniform noise, and then rescale them to $[0, 1]$. We transform the rescaled pixel values into logit space by $\mathbf{x} \mapsto \mathrm{logit}(\lambda + (1 - 2\lambda)\mathbf{x})$, where $\lambda = 10^{-6}$ for MNIST and $\lambda = 0.05$ for CIFAR-10, and perform density estimation in that space. In the case of CIFAR-10, we also augment the train set with horizontal flips of all train examples (as also done by Dinh et al. [4]).

Table 2 shows the results on MNIST and CIFAR-10. The performance of a class-conditional Gaussian is reported as a baseline for the conditional case. Log likelihoods are calculated in logit space. For unconditional density estimation, MADE MoG is the best performing model on MNIST, whereas MAF is the best performing model on CIFAR-10. For conditional density estimation, MAF is by far the best performing model on both datasets. On CIFAR-10, both MADE and MADE MoG performed significantly worse than the Gaussian baseline. MAF outperforms Real NVP in all cases.

The conditional performance of MAF is particularly impressive. MAF performs almost twice as well compared to its unconditional version and to every other model's conditional version. To facilitate comparison with the literature, Section E of the supplementary material reports results in bits/pixel. MAF (5) and MAF (10), the two best performing conditional models, achieve 3.02 and 2.98 bits/pixel respectively on CIFAR-10. This result is very close to the state-of-the-art 2.94 bits/pixel achieved by a conditional PixelCNN++ [27], even though, unlike PixelCNN++, our version of MAF does not incorporate prior image knowledge, and it pays a price for doing density estimation in a transformed real-valued space (PixelCNN++ directly models discrete pixel values).

## 5 Discussion

We showed that we can improve MADE by modelling the density of its internal random numbers. Alternatively, MADE can be improved by increasing the flexibility of its conditionals. The comparison between MAF and MADE MoG showed that the best approach is dataset specific; in our experiments MAF outperformed MADE MoG in 6 out of 9 cases, which is strong evidence of its competitiveness. MADE MoG is a universal density approximator; with sufficiently many hidden units and Gaussian components, it can approximate any continuous density arbitrarily well. It is an open question whether MAF with a Gaussian base density has a similar property (MAF MoG clearly does).

We also showed that the coupling layer used in Real NVP is a special case of the autoregressive layer used in MAF. In fact, MAF outperformed Real NVP in all our experiments. Real NVP has achieved impressive performance in image modelling by incorporating knowledge about image structure. Our results suggest that replacing coupling layers with autoregressive layers in the original version of Real NVP is a promising direction for further improving its performance. Real NVP maintains however the advantage over MAF (and autoregressive models in general) that samples from the model can be generated efficiently in parallel.

MAF achieved impressive results in conditional density estimation. Whereas almost all models we considered benefited from the additional information supplied by the labels, MAF nearly doubled its performance, coming close to state-of-the-art models for image modelling without incorporating

any prior image knowledge. The ability of MAF to benefit significantly from conditional knowledge suggests that automatic discovery of conditional structure (e.g. finding labels by clustering) could be a promising direction for improving unconditional density estimation in general.

Density estimation is one of several types of generative modelling, with the focus on obtaining accurate densities. However, we know that accurate densities do not necessarily imply good performance in other tasks, such as in data generation [31]. Alternative approaches to generative modelling include variational autoencoders [12, 25], which are capable of efficient inference of their (potentially interpretable) latent space, and generative adversarial networks [7], which are capable of high quality data generation. Choice of method should be informed by whether the application at hand calls for accurate densities, latent space inference or high quality samples. Masked Autoregressive Flow is a contribution towards the first of these goals.

**Acknowledgments**

We thank Maria Gorinova for useful comments. George Papamakarios and Theo Pavlakou were supported by the Centre for Doctoral Training in Data Science, funded by EPSRC (grant EP/L016427/1) and the University of Edinburgh. George Papamakarios was also supported by Microsoft Research through its PhD Scholarship Programme.

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
