[Supplementary Material]

# Supplementary material for "Masked Autoregressive Flow for Density Estimation"

**George Papamakarios**
University of Edinburgh
g.papamakarios@ed.ac.uk

**Theo Pavlakou**
University of Edinburgh
theo.pavlakou@ed.ac.uk

**Iain Murray**
University of Edinburgh
i.murray@ed.ac.uk

## A  Equivalence between MAF and IAF

In this section, we present the equivalence between MAF and IAF in full mathematical detail. Let $\pi_x(\mathbf{x})$ be the true density the train data $\{\mathbf{x}_n\}$ is sampled from. Suppose we have a MAF whose base density is $\pi_u(\mathbf{u})$, and whose transformation from $\mathbf{u}$ to $\mathbf{x}$ is $f$. The MAF defines the following density over the $\mathbf{x}$ space:

$$p_x(\mathbf{x}) = \pi_u\big(f^{-1}(\mathbf{x})\big)\left|\det\left(\frac{\partial f^{-1}}{\partial \mathbf{x}}\right)\right|. \tag{1}$$

Using the definition of $p_x(\mathbf{x})$ in Equation (1), we can write the Kullback–Leibler divergence from $\pi_x(\mathbf{x})$ to $p_x(\mathbf{x})$ as follows:

$$D_{\mathrm{KL}}(\pi_x(\mathbf{x})\,\|\,p_x(\mathbf{x})) = \mathbb{E}_{\pi_x(\mathbf{x})}(\log \pi_x(\mathbf{x}) - \log p_x(\mathbf{x})) \tag{2}$$

$$= \mathbb{E}_{\pi_x(\mathbf{x})}\left(\log \pi_x(\mathbf{x}) - \log \pi_u\big(f^{-1}(\mathbf{x})\big) - \log\left|\det\left(\frac{\partial f^{-1}}{\partial \mathbf{x}}\right)\right|\right). \tag{3}$$

The inverse transformation $f^{-1}$ from $\mathbf{x}$ to $\mathbf{u}$ can be seen as describing an implicit IAF with base density $\pi_x(\mathbf{x})$, which would define the following density over the $\mathbf{u}$ space:

$$p_u(\mathbf{u}) = \pi_x(f(\mathbf{u}))\left|\det\left(\frac{\partial f}{\partial \mathbf{u}}\right)\right|. \tag{4}$$

By making the change of variables $\mathbf{x} \mapsto \mathbf{u}$ in Equation (3) and using the definition of $p_u(\mathbf{u})$ in Equation (4) we obtain

$$D_{\mathrm{KL}}(\pi_x(\mathbf{x})\,\|\,p_x(\mathbf{x})) = \mathbb{E}_{p_u(\mathbf{u})}\left(\log \pi_x(f(\mathbf{u})) - \log \pi_u(\mathbf{u}) + \log\left|\det\left(\frac{\partial f}{\partial \mathbf{u}}\right)\right|\right) \tag{5}$$

$$= \mathbb{E}_{p_u(\mathbf{u})}(\log p_u(\mathbf{u}) - \log \pi_u(\mathbf{u})). \tag{6}$$

Equation (6) is the definition of the KL divergence from $p_u(\mathbf{u})$ to $\pi_u(\mathbf{u})$, hence

$$D_{\mathrm{KL}}(\pi_x(\mathbf{x})\,\|\,p_x(\mathbf{x})) = D_{\mathrm{KL}}(p_u(\mathbf{u})\,\|\,\pi_u(\mathbf{u})). \tag{7}$$

Suppose now that we wish to fit the implicit density $p_u(\mathbf{u})$ to the base density $\pi_u(\mathbf{u})$ by minimizing the above KL. This corresponds exactly to the objective minimized when employing IAF as a recognition network in stochastic variational inference [7], where $\pi_u(\mathbf{u})$ would be the (typically intractable) posterior. The first step in stochastic variational inference would be to rewrite the expectation in Equation (6) with respect to the base distribution $\pi_x(\mathbf{x})$ used by IAF, which corresponds exactly to Equation (3). This is often referred to as the reparameterization trick [6, 8]. The second step would be to approximate Equation (3) with Monte Carlo, using samples $\{\mathbf{x}_n\}$ drawn from $\pi_x(\mathbf{x})$, as follows:

$$D_{\mathrm{KL}}(p_u(\mathbf{u})\,\|\,\pi_u(\mathbf{u})) = \mathbb{E}_{\pi_x(\mathbf{x})}\left(\log \pi_x(\mathbf{x}) - \log \pi_u\big(f^{-1}(\mathbf{x})\big) - \log\left|\det\left(\frac{\partial f^{-1}}{\partial \mathbf{x}}\right)\right|\right) \tag{8}$$

$$\approx \frac{1}{N}\sum_n\left(\log \pi_x(\mathbf{x}_n) - \log \pi_u\big(f^{-1}(\mathbf{x}_n)\big) - \log\left|\det\left(\frac{\partial f^{-1}}{\partial \mathbf{x}}\right)\right|\right). \tag{9}$$

Using the definition of $p_x(\mathbf{x})$ in Equation (1), we can rewrite Equation (9) as

$$\frac{1}{N} \sum_n (\log \pi_x(\mathbf{x}_n) - \log p_x(\mathbf{x}_n)) = -\frac{1}{N} \sum_n \log p_x(\mathbf{x}_n) + \text{const.} \tag{10}$$

Since samples $\{\mathbf{x}_n\}$ drawn from $\pi_x(\mathbf{x})$ correspond precisely to the train data for MAF, we can recognize in Equation (10) the training objective for MAF. In conclusion, training a MAF by maximizing its total log likelihood $\sum_n \log p_x(\mathbf{x}_n)$ on train data $\{\mathbf{x}_n\}$ is equivalent to variationally training an implicit IAF with MAF's base distribution $\pi_u(\mathbf{u})$ as its target.

## B    Batch normalization

In our implementation of MAF, we inserted a batch normalization layer [5] between every two autoregressive layers, and between the last autoregressive layer and the base distribution. We did the same for Real NVP (the original implementation of Real NVP also uses batch normalization layers between coupling layers [3]). The purpose of a batch normalization layer is to normalize its inputs $\mathbf{x}$ to have approximately zero mean and unit variance. In this section, we describe in full detail our implementation of batch normalization and its use as a layer in normalizing flows.

A batch normalization layer can be thought of as a transformation between two vectors of the same dimensionality. For consistency with our notation for autoregressive and coupling layers, let $\mathbf{x}$ be the vector closer to the data, and $\mathbf{u}$ be the vector closer to the base distribution. Batch normalization implements the transformation $\mathbf{x} = f(\mathbf{u})$ defined by

$$\mathbf{x} = (\mathbf{u} - \boldsymbol{\beta}) \odot \exp(-\boldsymbol{\gamma}) \odot (\mathbf{v} + \epsilon)^{\frac{1}{2}} + \mathbf{m}. \tag{11}$$

In the above, $\odot$ denotes elementwise multiplication. All other operations are to be understood elementwise. The inverse transformation $f^{-1}$ is given by

$$\mathbf{u} = (\mathbf{x} - \mathbf{m}) \odot (\mathbf{v} + \epsilon)^{-\frac{1}{2}} \odot \exp \boldsymbol{\gamma} + \boldsymbol{\beta}, \tag{12}$$

and the absolute determinant of its Jacobian is

$$\left| \det \left( \frac{\partial f^{-1}}{\partial \mathbf{x}} \right) \right| = \exp \left( \sum_i \left( \gamma_i - \frac{1}{2} \log(v_i + \epsilon) \right) \right). \tag{13}$$

Vectors $\boldsymbol{\beta}$ and $\boldsymbol{\gamma}$ are parameters of the transformation that are learnt during training. In typical implementations of batch normalization, parameter $\boldsymbol{\gamma}$ is not exponentiated. In our implementation, we chose to exponentiate $\boldsymbol{\gamma}$ in order to ensure its positivity and simplify the expression of the log absolute determinant. Parameters $\mathbf{m}$ and $\mathbf{v}$ correspond to the mean and variance of $\mathbf{x}$ respectively. During training, we set $\mathbf{m}$ and $\mathbf{v}$ equal to the sample mean and variance of the current minibatch (we used minibatches of 100 examples). At validation and test time, we set them equal to the sample mean and variance of the entire train set. Other implementations use averages over minibatches [5] or maintain running averages during training [3]. Finally, $\epsilon$ is a hyperparameter that ensures numerical stability if any of the elements of $\mathbf{v}$ is near zero. In our experiments, we used $\epsilon = 10^{-5}$.

## C    Number of parameters

To get a better idea of the computational trade-offs between different model choices versus the performance gains they achieve, we compare the number of parameters for each model. We only count connection weights, as they contribute the most, and ignore biases and batch normalization parameters. We assume that masking reduces the number of connections by approximately half.

For all models, let $D$ be the number of inputs, $H$ be the number of units in a hidden layer and $L$ be the number of hidden layers. We assume that all hidden layers have the same number of units (as we did in our experiments). For MAF MoG, let $C$ be the number of components per conditional. For Real NVP and MAF, let $K$ be the number of coupling layers/autoregressive layers respectively. Table 1 lists the number of parameters for each model.

For each extra component we add to MADE MoG, we increase the number of parameters by $DH$. For each extra autoregressive layer we add to MAF, we increase the number of parameters by

Table 1: Approximate number of parameters for each model, as measured by number of connection weights. Biases and batch normalization parameters are ignored.

|  | # of parameters |
| --- | --- |
| MADE | $\frac{3}{2}DH + \frac{1}{2}(L-1)H^2$ |
| MADE MoG | $\left(C + \frac{1}{2}\right)DH + \frac{1}{2}(L-1)H^2$ |
| Real NVP | $2KDH + 2K(L-1)H^2$ |
| MAF | $\frac{3}{2}KDH + \frac{1}{2}K(L-1)H^2$ |

$\frac{3}{2}DH + \frac{1}{2}(L-1)H^2$. If we have one or two hidden layers $L$ (as we did in our experiments) and assume that $D$ is comparable to $H$, the number of extra parameters in both cases is about the same. In other words, increasing flexibility by stacking has a parameter cost that is similar to adding more components to the conditionals, as long as the number of hidden layers is small.

Comparing Real NVP with MAF, we can see that Real NVP has about $1.3$ to $2$ times more parameters than a MAF of comparable size. Given that our experiments show that Real NVP is less flexible than a MAF of comparable size, we can conclude that MAF makes better use of its available capacity. The number of parameters of Real NVP could be reduced by tying weights between the scaling and shifting networks.

## D  Additional experimental details

### D.1  Models

MADE, MADE MoG and each autoregressive layer in MAF is a feedforward neural network (with masked weight matrices), with $L$ hidden layers of $H$ hidden units each. Similarly, each coupling layer in Real NVP contains two feedforward neural networks, one for scaling and one for shifting, each of which also has $L$ hidden layers of $H$ hidden units each. For each dataset, we gave a number of options for $L$ and $H$ (the same options where given to all models) and for each model we selected the option that performed best on the validation set. Table 2 lists the combinations of $L$ and $H$ that were given as options for each dataset.

In terms of nonlinearity for the hidden units, MADE, MADE MoG and MAF used rectified linear units, except for the GAS datasets where we used hyperbolic tangent units. In the coupling layer of Real NVP, we used hyberbolic tangent hidden units for the scaling network and rectified linear hidden units for the shifting network.

Table 2: Number of hidden layers $L$ and number of hidden units $H$ given as options for each dataset. Each combination is reported in the format $L \times H$.

| POWER | GAS | HEPMASS | MINIBOONE | BSDS300 | MNIST | CIFAR-10 |
| --- | --- | --- | --- | --- | --- | --- |
| $1 \times 100$ | $1 \times 100$ | $1 \times 512$ | $1 \times 512$ | $1 \times 512$ | $1 \times 1024$ | $1 \times 1024$ |
| $2 \times 100$ | $2 \times 100$ | $2 \times 512$ | $2 \times 512$ | $2 \times 512$ | | $2 \times 1024$ |
| | | | | $1 \times 1024$ | | $2 \times 2048$ |
| | | | | $2 \times 1024$ | | |

### D.2  Datasets

In the following paragraphs, we give a brief description of the four UCI datasets (POWER, GAS, HEPMASS, MINIBOONE) and of the way they were preprocessed.

**POWER**. The POWER dataset [1] contains measurements of electric power consumption in a household over a period of $47$ months. It is actually a time series but was treated as if each example

were an i.i.d. sample from the marginal distribution. The time feature was turned into an integer for the number of minutes in the day and then uniform random noise was added to it. The date was discarded, along with the global reactive power parameter, which seemed to have many values at exactly zero, which could have caused arbitrarily large spikes in the learnt distribution. Uniform random noise was added to each feature in the interval $[0, \epsilon_i]$, where $\epsilon_i$ is large enough to ensure that with high probability there are no identical values for the $i^{\text{th}}$ feature but small enough to not change the data values significantly.

**GAS**. Created by Fonollosa et al. [4], this dataset represents the readings of an array of 16 chemical sensors exposed to gas mixtures over a 12 hour period. Similarly to POWER, it is a time series but was treated as if each example were an i.i.d. sample from the marginal distribution. Only the data from the file `ethylene_CO.txt` was used, which corresponds to a mixture of ethylene and carbon monoxide. After removing strongly correlated attributes, the dimensionality was reduced to 8.

**HEPMASS**. Used by Baldi et al. [2], this dataset describes particle collisions in high energy physics. Half of the data are examples of particle-producing collisions (positive), whereas the rest come from a background source (negative). Here we used the positive examples from the "1000" dataset, where the particle mass is 1000. Five features were removed because they had too many reoccurring values; values that repeat too often can result in spikes in the density and misleading results.

**MINIBOONE**. Used by Roe et al. [9], this dataset comes from the MiniBooNE experiment at Fermilab. Similarly to HEPMASS, it contains a number of positive examples (electron neutrinos) and a number of negative examples (muon neutrinos). Here we use the positive examples. These had some obvious outliers (11) which had values at exactly $-1000$ for every column and were removed. Also, seven of the features had far too high a count for a particular value, e.g. $0.0$, so these were removed as well.

Table 3 lists the dimensionality and the number of train, validation and test examples for all seven datasets. The first three datasets in Table 3 were subsampled so that the product of the dimensionality and number of examples would be approximately 10M. For the four UCI datasets, 10% of the data was held out and used as test data and 10% of the remaining data was used as validation data. From the BSDS300 dataset we randomly extracted 1M patches for training, 50K patches for validation and 250K patches for testing. For MNIST and CIFAR-10 we held out 10% of the train data for validation. We augmented the CIFAR-10 train set with the horizontal flips of all remaining 45K train examples.

Table 3: Dimensionality $D$ and number of examples $N$ for each dataset.

|  | $D$ | $N$ train | validation | test |
|---|---|---|---|---|
| POWER | 6 | 1,659,917 | 184,435 | 204,928 |
| GAS | 8 | 852,174 | 94,685 | 105,206 |
| HEPMASS | 21 | 315,123 | 35,013 | 174,987 |
| MINIBOONE | 43 | 29,556 | 3,284 | 3,648 |
| BSDS300 | 63 | 1,000,000 | 50,000 | 250,000 |
| MNIST | 784 | 50,000 | 10,000 | 10,000 |
| CIFAR-10 | 3072 | 90,000 | 5,000 | 10,000 |

# E  Additional results

## E.1  Pairwise comparison

On the MINIBOONE dataset, the model with highest average test log likelihood is MAF MoG (5). However, due to the relatively small size of this dataset, the average test log likelihoods of some other models have overlapping error bars with that of MAF MoG (5). To assess whether the differences are statistically significant, we performed a pairwise comparison, which is a more powerful statistical test. In particular, we calculated the difference in test log probability between every other model and MAF MoG (5) on each test example, and assessed whether this difference is significantly positive,

Table 4: Pairwise comparison results for MINIBOONE. Values correspond to average difference in log probability (in nats) from the best performing model, i.e. MAF MoG (5). Error bars correspond to 2 standard deviations. Significantly positive values indicate that MAF MoG (5) performs better.

|  | MINIBOONE |
| --- | --- |
| Gaussian | $25.55 \pm 0.88$ |
| MADE | $3.91 \pm 0.20$ |
| MADE MoG | $0.59 \pm 0.16$ |
| Real NVP (5) | $1.87 \pm 0.16$ |
| Real NVP (10) | $2.15 \pm 0.21$ |
| MAF (5) | $\mathbf{0.07 \pm 0.11}$ |
| MAF (10) | $0.55 \pm 0.12$ |
| MAF MoG (5) | $\mathbf{0.00 \pm 0.00}$ |

which would indicate that MAF MoG (5) performs significantly better. The results of this comparison are shown in Table 4. We can see that MAF MoG (5) is significantly better than all other models except for MAF (5).

## E.2  Bits per pixel

In the main text, the results for MNIST and CIFAR-10 were reported in log likelihoods in logit space, since this is the objective that the models were trained to optimize. For comparison with other results in the literature, in Table 5 we report the same results in bits per pixel. For CIFAR-10, different colour components count as different pixels (i.e. an image is thought of as having $32 \times 32 \times 3$ pixels).

In order to calculate bits per pixel, we need to transform the densities returned by a model (which refer to logit space) back to image space in the range $[0, 256]$. Let $\mathbf{x}$ be an image of $D$ pixels in logit space and $\mathbf{z}$ be the corresponding image in $[0, 256]$ image space. The transformation from $\mathbf{z}$ to $\mathbf{x}$ is

$$\mathbf{x} = \text{logit}\left(\lambda + (1 - 2\lambda)\frac{\mathbf{z}}{256}\right), \tag{14}$$

where $\lambda = 10^{-6}$ for MNIST and $\lambda = 0.05$ for CIFAR-10. If $p(\mathbf{x})$ is the density in logit space as returned by the model, using the above transformation the density of $\mathbf{z}$ can be calculated as

$$p_z(\mathbf{z}) = p(\mathbf{x})\left(\frac{1 - 2\lambda}{256}\right)^D \left(\prod_i \sigma(x_i)(1 - \sigma(x_i))\right)^{-1}, \tag{15}$$

where $\sigma(\cdot)$ is the logistic sigmoid function. From that, we can calculate the bits per pixel $b(\mathbf{x})$ of image $\mathbf{x}$ as follows:

$$b(\mathbf{x}) = -\frac{\log_2 p_z(\mathbf{z})}{D} \tag{16}$$

$$= -\frac{\log p(\mathbf{x})}{D \log 2} - \log_2(1 - 2\lambda) + 8 + \frac{1}{D}\sum_i(\log_2 \sigma(x_i) + \log_2(1 - \sigma(x_i))). \tag{17}$$

The above equation was used to convert between the average log likelihoods reported in the main text and the results of Table 5.

## E.3  Generated images

Figures 1, 2 and 3 show generated images and real examples for BSDS300, MNIST and CIFAR-10 respectively. Images were generated by the best performing model for each dataset, which is MAF MoG (5) for BSDS300, conditional MAF (5) for MNIST and conditional MAF (10) for CIFAR-10.

The BSDS300 generated images are visually indistinguishable from the real ones. For MNIST and CIFAR-10, generated images lack the fidelity produced by modern image-based generative approaches, such as RealNVP [3] or PixelCNN++ [10]. This is because our version of MAF has

Table 5: Bits per pixel for conditional density estimation (lower is better). The best performing model for each dataset is shown in bold. Error bars correspond to 2 standard deviations.

| | MNIST | | CIFAR-10 | |
|---|---|---|---|---|
| | unconditional | conditional | unconditional | conditional |
| Gaussian | $2.01 \pm 0.01$ | $1.97 \pm 0.01$ | $4.63 \pm 0.01$ | $4.79 \pm 0.02$ |
| MADE | $2.04 \pm 0.01$ | $2.00 \pm 0.01$ | $5.67 \pm 0.01$ | $5.65 \pm 0.01$ |
| MADE MoG | $\mathbf{1.41 \pm 0.01}$ | $1.39 \pm 0.01$ | $5.93 \pm 0.01$ | $5.80 \pm 0.01$ |
| Real NVP (5) | $1.93 \pm 0.01$ | $1.94 \pm 0.01$ | $4.53 \pm 0.01$ | $4.50 \pm 0.01$ |
| Real NVP (10) | $2.02 \pm 0.02$ | $2.02 \pm 0.08$ | $4.54 \pm 0.01$ | $4.58 \pm 0.01$ |
| MAF (5) | $1.89 \pm 0.01$ | $\mathbf{0.59 \pm 0.01}$ | $4.36 \pm 0.01$ | $3.02 \pm 0.01$ |
| MAF (10) | $1.91 \pm 0.01$ | $0.61 \pm 0.01$ | $\mathbf{4.31 \pm 0.01}$ | $\mathbf{2.98 \pm 0.01}$ |
| MAF MoG (5) | $1.52 \pm 0.01$ | $1.51 \pm 0.01$ | $4.37 \pm 0.01$ | $4.36 \pm 0.01$ |

(a) Generated images      (b) Real images

Figure 1: Generated and real images from BSDS300.

no knowledge about image structure, as it was designed for general-purpose density estimation and not for realistic-looking image synthesis. However, if the latter is desired, it would be possible to incorporate image modelling techniques in the design of MAF (such as convolutions or a multi-scale architecture as used by Real NVP [3]) in order to improve quality of generated images.