[Reviews · NeurIPS 2017]

Reviewer 1



This is a really well written paper that presents a unified view of various state-of-the-art techniques for density modeling. More specifically: the authors show that autoregressive models, normalizing flows, inverse autoregressive flows and RealNVP/NICE can be easily understood from a common perspective. Some aspects of this perspective were already known and, for example, described in the IAF paper by Kingma et. al. But this paper describes the connections between these models and techniques in far more detail and in a plausible and easy to follow way. Furthermore, starting from this perspective, the authors derive a new variant and show that they obtain very promising results that compete with (or beat) various state of the art approaches.

Reviewer 2



SUMMARY * The paper presents a method for density estimation based on a stack of autoregressive models, exploiting a normalizing flow interpretation of autoregressive models. This is an approach to increase the flexibility of the conditionals in masked autoregressive density estimation in contrast to say mixture models. CLARITY * The paper is very well written. It does a good job at explaining the relations of existing methods. * The experiments are presented in a convincing way and code is provided. RELEVANCE * The proposed approach appears sensible and seems to provide substantial benefits over other approaches, especially in the case of conditional density estimation. * The experiments show a good performance of the proposed methods. While other methods have shown very good performance when incorporating knowledge about the problem domain, the proposed methods are demonstrated to provide a competitive alternative for general purpose density estimation. * The discussion seems to open promising avenues for further research.

Reviewer 3



This paper is generally well written and I enjoy reading it. It introduces an expressive density model called masked autoregressive flow (MAF) that stacks multiple MADE layers to form a normalizing flow. Although it seems a bit incremental since the techniques involved have been studied in IAF and MADE, this paper does a good job elaborating on different types of generative modeling and providing guidelines for their use cases. It also makes a connection between MAF and IAF. Only a few comments/questions below: * It'd be helpful to motivate a bit more on the advantage of density models. What are the applications or downstream tasks that make density models more suitable than their alternatives such as VAEs or GANs? For example, does the proposed density model admit efficient algorithms for marginalization or other inference tasks? * When applying multiple transformations in the normalizing flow, is it implied that the conditionals need not be Gaussian as in eq. 2 because the density is computed according to the density of u? When u is transformed from some lower-level random vectors, it could very likely be non-Gaussian. Do I understand correctly? * Does the base density of an MAF (line 225) refer to the density of u in eq. 3 or the density of p(x_i|x_{1:i-1}) in eq. 1? I assumed it's the former. * When comparing the results from MAF (5), MAF (10) and MAF MoG (5) on the data sets POWER and MNIST, MAF MoG outperforms the other two. I have the impression that if using multiple normalizing flow layers where each layer has multiple non-linear MADE layers, MAF should be universal without MoG. What's the authors' opinion on this?